# Multi-Year Examination of School-Based Programs in Preventing Childhood Obesity: A Case of a State Policy in the U.S.

**DOI:** 10.3390/ijerph17249425

**Published:** 2020-12-16

**Authors:** Chang-Yong Jang, Nam-Gyeong Gim, Yoonhee Kim, TaeEung Kim

**Affiliations:** 1Korea Institute of Sport Science, Seoul 01794, Korea; jangcy529@kspo.or.kr; 2Department of Administration, Yuk-buk Elementary School, Yongin 17061, Korea; v-ness@daum.net; 3Graduate School of Education, Soonchunhyang University, Asan 31538, Korea; kyhee99@sch.ac.kr; 4Department of Epidemiology, University of California, Irvine, CA 92697, USA

**Keywords:** physical activity, physical education, nutrition, childhood obesity

## Abstract

This study examined the association between the obesogenic factors and the risk of suffering from weight excess in school-based state programs regarding physical activity, physical education, nutrition standards, and nutrition education in preventing childhood obesity. Data were drawn from the 1999–2011 Youth Risk Behavior Survey in the State of Mississippi (N = 8862; grades 9–12). Logistic regression with year-fixed effects was performed to capture the influence of the legislation on teenage obesity, controlling for demographics and nutrition- and physical activity-related behaviors. The age-, sex-, and ethnicity-adjusted mean of the body mass index had reduced since 2007 (year 1999: 23.52; year 2001: 23.53; year 2003: 23.76; year 2007: 24.26; year 2009: 24.29; and year 2011: 23.91). The legislation was significantly associated with a decreased likelihood of being overweight (year 2007, odds ratio (OR) = 0.686; year 2009, OR = 0.739; and year 2011, OR = 0.760; all *p* < 0.01). Children who were more sedentary, more frequently fasted to lose weight, and were less physically active and likelier to be overweight (OR = 1.05, 1.37, and 0.97, respectively; all *p* < 0.05), as were African-American children (OR = 0.64; *p* < 0.05) and female students (OR = 1.59; *p* < 0.05). In conclusion, schools are among the most easily modifiable settings for preventing childhood obesity and reducing its prevalence, with the implementation of physical activity and nutritional policies.

## 1. Introduction

The high prevalence of overweight and obesity among children and adolescents has been well-recognized in the United States (US). Obesity is among the top non-transmittable chronic diseases, and accounts for 60% of deaths in the US [1]. Childhood obesity is defined as an age- and sex-specific body mass index (BMI) at or above the 95th percentile [2]. In the US, there has been a dramatic increase in the proportion of children and adolescents who are overweight or obese, over the past few decades [3]. More than 10% of students aged 5–17 years were overweight in 2004 [4]. The number of overweight children in this age group has been increasing by 0.5%, per year in the US, over the last two decades. The prevalence of childhood obesity has doubled among children aged 6–11 years, and tripled among adolescents aged 12–19 years [4,5]. It is extremely important for children to maintain a healthy weight by forming healthy habits such as eating more fruits and vegetables, and being more active and less sedentary.

Childhood obesity has huge detrimental impacts on an individual’s physical [6] and mental [7] health. It can lead to impaired glucose tolerance [8,9], cardiovascular diseases in adulthood [10], eating disorder behaviors including fasting and vomiting [11,12], insulin resistance, and abnormal lipid accumulation [10], multiple cardiovascular risks and atherosclerosis [13], and low self-esteem and depression in life [14]. Moreover, childhood obesity has negative effects on overall physical functioning and health outcomes in adulthood [15,16]; it can cause cancers of the esophagus, colon, rectum, liver, gallbladder, pancreas, and kidney [17] and increase mortality [18,19]. Therefore, implementing effective intervention programs or policies to reduce childhood obesity is vital.

In the last decade, many states such as Mississippi and Louisiana have proposed and enacted school-based state policies to reduce the prevalence of childhood obesity. One of the significant school-based state policies aimed at reducing the prevalence of childhood obesity, in Mississippi, was implemented in 2007. The state legislation implemented the policy in three steps: First, it implemented school nutrition and wellness programs that regulated, marketed, and prepared healthy food and beverages for students and staff, for breakfast and lunch, and increased their participation in child nutrition and school breakfast and lunch programs. Second, it enhanced school nutrition education by conducting programs focused on promoting physical activity and healthy eating habits, and abstaining from tobacco and illegal drugs. Third, it adopted national guidelines recommending participation in physical activity for 150 min per week and health education for 45 min per week [20]. However, few studies have evaluated the effectiveness of these school-based state policies. Evaluating past and ongoing policy interventions can prove helpful for practitioners, stakeholders, and policy-makers in developing more reliable and feasible health-behavior interventions [21].

The purpose of this study was to identify the association between the obesogenic factors and the risk of suffering from weight excess in school-based state policies to reduce the prevalence of obesity among children in a State in the US. This study is a multi-year cross-sectional study that used the Bronfenbrenner’s ecological system theory [22,23] as a conceptual framework to evaluate if school-based state policies influence obesity prevention behaviors, such as eating healthy and participating in physical activity [24], in adolescents, in the State of Mississippi. In this model, one level of the ecological system could affect an individual’s behaviors as well as one’s development, directly or indirectly, through a multilevel contextual mechanism [22,23]. In other words, a student nested in the center of an ecological system comprising a microsystem (e.g., schools, in our study), mesosystem (e.g., community), ecosystem (e.g., the state level), and macrosystem (e.g., customs and laws). As shown in Figure 1, school-aged children’s ecological environments encompass schools, which are, in turn, part of larger social environments such as entire states [25].

It is hypothesized that (1) school nutrition policies have increased students’ consumption of favorable nutrition (e.g., fruits and vegetables); (2) school policies on physical activity have increased the level of students’ physical activity and decreased the amount of sedentary behavior; (3) school-based nutrition and physical activity policies have discouraged students from resorting to unhealthy behaviors to lose weight (e.g., fasting and vomiting); (4) school-based nutrition and physical activity policies have had a positive effect on decreasing the prevalence of childhood obesity.

## 2. Materials and Methods

### 2.1. Study Population and Sampling

This study used the Youth Risk Behavior Surveillance System (YRBSS) [2], a school-based and self-reported survey, representing a sample of public and private middle or high-school students in the US. Data from 1999, 2001, 2003, 2007, 2009, and 2011 were used in this study. Data of 2005 were omitted due to the unavailability of information on a few key variables. The CDC conducts this nation-wide survey with both public and private high-school students, who are in grades 9–12, from 50 states, every other year, to monitor health-risk behaviors and the prevalence of obesity and asthma among the youth [2]. The data of this study were only data from Mississippi State.

This study was a multi-year retrospective cross-sectional study. In order to evaluate the effectiveness of the state legislation implemented in 2007, the years prior to 2007 were considered as the baseline period of the research, and the post-policy assessment was conducted using the data of years 2009 and 2011. Therefore, it is to investigate how obesity rates in children prior to 2007, namely those in 1999, 2001, and 2003, and those in 2009 and 2011 after 2007, were changed at the beginning of the less effective policy. This study was exempted by the Institutional Review Board at the Indiana University’ institution.

### 2.2. Measurement 

The outcome variable of this study was the BMI percentile. BMI was calculated as weight in kilograms divided by square of height in meters; BMI < 85th percentile was considered healthy weight and BMI ≥ 85th percentile was considered overweight or obese, based on sex and BMI-for-age of children [26].

The covariates of this study included individual-level demographics and school-level health behavior-related variables, which were measured as follows. First, demographic variables included age, ethnicity (recoded as whites, blacks, and others) and sex. Secondly, school-level health behavior-related variables included the level of abusive behavior for weight control, the level of consumption of fruits and vegetables, and the level of physical activity and sedentary behavior. Abusive behavior for weight control was measured based on the binary scale (yes or no) recoded from three X-point survey items; in the past 30 days, having not eaten for 24 h or more; taking any diet pills, powders, or liquids without a doctor’s advice; and vomiting or taking laxatives to lose weight. These three items were summed as a composite score which determined the presence of abusive behavior for weight control, if the score ranged from 3 to 6. The consumption of fruits and vegetables was measured on seven 7-point Likert scales (e.g., in the past 7 days, how many times an individual consumed 100% fruits juices, fruits, green salads, potatoes, carrots and other vegetables). A composite score was obtained by summing all seven items (range 7–42). Physical activity was assessed using two 6-point scales and one 4-point Likert scale (e.g., how many days, in a week, did students attend physical education classes week when they were in school, how many minutes did they spend actually exercising or playing sports during an average physical education (PE) class and how many of them played team sports in the past 12 months). A composite scale score was used, which summed each item, for scores ranging between 2 and 10. Finally, students’ watching TV were assessed by one item with a 7-point Likert scale: how many hours students watched TV on an average school day (score range 1 to 7).

### 2.3. Data Analyses

Descriptive statistics such as means, standard deviations, or percentage were reported to describe the study sample’s demographic characteristics, such as age, sex, ethnicity, and BMI. *t*-tests and chi-square tests were performed to identify any differences in the consumption of fruits, vegetables and nutrition, physical activity, watching TV, and abusive behavior to lose weight, by sex, ethnicity, and age. We also performed factorial ANOVAs to examine the trends of the four nutrition- and physical activity-related variables, by year. 

A logistic regression was used to predict overweight status (i.e., normal weight vs. overweight/obese) among middle or high-school students, in Mississippi, controlling for covariates including age, sex, ethnicity, the level of consumption of fruits and vegetables, nutrition, physical activity, sedentary behavior, abusive behavior to lose weight, and year. This analysis is useful for predicting a discrete outcome such as group membership from a set of variables that may be continuous or mix [27]. SAS (version 9.3) was used to conduct all statistical analyses. Statistical tests were conducted using a 0.05 alpha level and a 95% confidence interval (CI).

## 3. Results

In total, 8,862 children were identified (mean age = 16.21; SD = 1.23 years; girls = 53.80%). The age-, sex-, and ethnicity-adjusted mean of the body mass index had reduced since 2007 (year 1999: 23.52; year 2001: 23.53; year 2003: 23.76; year 2007: 24.26; year 2009: 24.29; and year 2011: 23.91) (Figure 2). The ethnicity of the participants was quite diverse, with most of them being white (46.5%) and black (46.7%). Their grades were evenly distributed; 29.0% of them were in grade 9, 24.5% in grade 12, 24.1% in grade 11, and 21.7% in grade 10. The four independent-sample *t*-tests, by sex, as shown in Table 1, revealed significant differences in abusive behavior to lose weight, consumption of fruits and vegetables (*p* < 0.01), and physical activity (all *p* < 0.01). However, in terms of sedentary behavior, there was no significant difference between male and female students (*p* = 0.07).

In the Table 1, significant differences in the independent variables, based on four grade levels (e.g., grades 9, 10, 11, and 12) in terms of the four variables (e.g., abusive behavior to lose weight, consumption of fruits and vegetables, physical activity, and sedentary behavior) showed that only two variables displayed statistically significant differences-sedentary behavior (*p* < 0.01) and physical activity (*p* < 0.01). However, in terms of the consumption of fruits and vegetables (*p* = 0.09) and abusive behavior to lose weight (*p* = 0.43), there were no significant differences between 9th grade to 12th grade. As shown in the Table 2, factorial ANOVAs indicated significant differences in the consumption of fruits and vegetables (*p* < 0.01), sedentary behavior (*p* < 0.01), physical activity (*p* < 0.01), and abusive behavior to lose weight (*p* < 0.01) between 2007 and 2011. Moreover, regarding the year-related categorical variables in the Table 2, significant differences in the independent variables, by year (e.g., 1999, 2001, 2003, 2007, 2009, and 2011) with the four variables (e.g., abusive behavior to lose weight, consumption of fruits and vegetables, physical activity, and sedentary behavior), were observed for the consumption of fruits and vegetables (*p* < 0.01), sedentary behavior (*p* < 0.01), physical activity (*p* < 0.01), and abusive behavior to lose weight (*p* < 0.01) (see Figure 3 & Table 2).

### Adolescents’ Obesity and Multivariable Logistic Regression

The age-, sex-, and ethnicity-adjusted mean of the BMI had decreased, over time, since 2007 (year 1999 = 23.52; year 2001 = 23.53; year 2003 = 23.76; year 2007 = 24.26; year 2009 = 24.29; and year 2011 = 23.91). The odds ratios (ORs) and the 95% CIs of the covariates are shown in Table 3. The four independent continuous variables associated with adolescents’ school-based intervention policies were significantly different from the ORs in adolescents’ binary BMI levels (i.e., overweight (≥85th percentile) vs. healthy weight (<85th percentile) (see Table 3). First, for every one-unit (1 h) increase in the level of engagement in watching TV the likelihood of being overweight was 4.6% times higher (OR = 1.05, 95% CIs = 1.02–1.07). Second, for every one-unit increase in the level of physical activity, the odds of being overweight decreased by 2.7% (OR = 0.97, 95% CIs = 0.96–0.99). Third, for every one-unit increase in the level of abusive behavior to lose weight, the odds of being overweight increased by 37.3% (OR = 1.37, 95% CIs = 1.29–1.46). Finally, for every one-unit (one year) increase in age, the odds of being overweight decreased by 8.1% (OR = 0.92, 95% CIs = 0.88–0.95).

In addition, three categorical variables, regarding adolescents’ school-based intervention policies, displayed significantly different ORs for adolescents’ binary BMI levels as shown in Table 3. First, on comparing the odds of being overweight, African-American students were 36% less likely to be overweight (OR = 0.64, 95% CIs = 0.58–0.71) than white students. However, no significant differences were observed between white students and those of other ethnicity. Second, female students had 56% greater adjusted odds of being overweight (OR = 1.59, 95% CIs = 1.44–1.74) compared to male students. School-based childhood obesity policies have been implemented since 2007. Based on the fixed-year effects (Figure 3 and Table 3), the odds of being overweight decreased from 1999 (year 2001, OR = 0.95, 95% CIs = 0.81–1.12; year 2003, OR = 0.85, 95% CIs = 0.72–1.01; year 2007, OR = 0.69, 95% CIs = 0.58–0.81; year 2009, OR = 0.74, 95% CIs = 0.63–0.87; and year 2011, OR = 0.76, 95% CIs = 0.65–0.89)

## 4. Discussion

This was a multi-year cross-sectional study that aimed to examine the association between the obesogenic factors and the risk of suffering from weight excess in school policies of state legislation regarding childhood obesity (such as nutrition standards, nutrition education, and physical activity) in Mississippi, US. Based on the results of this study, the state legislation on school policies regarding childhood obesity has, overall, been effective since 2007. Furthermore, this study supported the theory that schools, as community-based intervention programs that regulate child weight and fitness, could successfully increase fitness levels and decrease the prevalence of obesity [28,29]. In addition, this study showed that school-based nutrition and physical education policies could encourage adolescents to reduce engagement in abusive behaviors to lose weight. It was reviewed the effectiveness of school-based childhood obesity intervention programs, in terms of dietary intake and physical activity levels, and recommended that a school-based prevention obesity intervention program, based on a combination of nutrition intake and physical activity, may prevent a child from becoming overweight [30].

This study also revealed that school nutrition policies could increase students’ consumption of favorable nutrition such as fruits and vegetables. This positive state legislation effect was concluded in other states such as Texas and Utah. The Texas Public School Nutrition Policy intervention program among middle-school students [31], for lunchtime food consumption, was effective in encouraging students to eat more vegetables, milk, and other nutrients (such as protein, fiber, vitamins, calcium, and sodium) and a lower number of less-favorable items (e.g., sugared beverages and snacks). In addition, the School Wellness Policies of Utah [32], including nutrition and physical activity, improved the school nutrition and physical activity environment.

Given the fact that childhood obesity has been shown to be influenced by inadequate dietary habits such as imbalances between energy intake and expenditure [33], insufficient fruit and vegetable consumption, and excessive intake of high-fat and high-caloric foods [34,35], the state legislation on school nutrition policies could play a role in obesity prevention. In particular, only one out of five high-school students eat five or more servings of fruits and vegetables, per day [36], and less than one-fourth of younger children consume the nutrients recommended by the federal guidelines [37]. It has also been hypothesized that school policies pertaining to childhood obesity increased students’ physical activity levels and decreased the engagement in sedentary behavior. [38] indicated that sedentary behaviors in childhood (e.g., watching TV, playing video games, and using the computer) should not exceed more than 2 h per day. Several other researchers have also stressed on the importance of discouraging physical inactivity and preventing sedentary lifestyles [39,40,41].

It is proven that schools can act as a modifiable environmental setting in the minimization and prevention of childhood obesity, through the utilization of physical activity and nutritional policies. Determining the factors that contribute to a decreased prevalence of childhood obesity, in such programs, will help educators facilitate engagement in school-based preventive obesity programs. This study reveals that school polices related to children’s eating and physical activity behaviors could have a huge influence in reducing the prevalence of childhood obesity.

Several interesting associations were observed. First, behavioral factors (i.e., sedentary lifestyles, physical activity, and abusive behaviors to lose weight) were proven to be significantly associated with the prevalence of obesity in adolescents. Engagement in abusive behaviors to lose weight should be paid attention to, in particular. Therefore, health practitioners and professionals should not only inform and educate students on the importance of positive behaviors, but also lay emphasis on how to substitute unhealthy weight management methods with healthier practices.

The second association of great interest is the fact that the consumption of fruits and vegetables was not significantly associated with the prevalence of adolescent obesity. In this study, the level of consumption of fruits and vegetables did not significantly help predict an adolescent’s BMI category. This could be attributed to the fact that, while an increased consumption of fruits and vegetables is not directly associated with the epidemic of obesity among school-aged adolescents, it could minimize the total consumption of high-fat and energy foods [34,35] and prevent the occurrence of cardiovascular disease and some cancers [42]. Therefore, policy-makers and/or stakeholders should not rule out those nutrition practices, despite the low cost–benefit ratio of medical or social interventions.

In the present study, girls were likelier to be overweight/obese than boys. In addition, black students were less likely to be overweight/obese than those belonging to other ethnicity; this could reflect differences in the regional and cultural backgrounds. This information should not only be used as a resource for interventions geared toward increasing participation in obesity management programs, but also to inform national health policy perspectives, including treatment and outcomes for school-based obesity programs.

Physical activity and nutrition policies are the most prominent environmental factors affecting adolescent obesity [43]. However, few schools implement well-structured physical activity programs and nutrition policies for students, due to the pressure associated with academic achievement, lack of financing, and poor-quality exercise programs [44]. Until 2007, Mississippi did not have any effective state legislation policy pertaining to childhood obesity, including both physical activity and nutrition, even though it has the highest rate of childhood obesity of all the states in the US [2]. The findings of this study show that the 2007 state legislation on school policies of childhood obesity was positively associated with healthier behaviors among students (e.g., increased physical activity and nutrition) and could potentially decrease obesity prevalence among adolescents.

Multiple factors are associated with adolescent obesity, including the excessive intake of unhealthy foods, lower levels of physical activity, and more sedentary lifestyles. Therefore, individuals’ efforts alone may not be sufficient in preventing childhood obesity; cooperation with social structures and policies is vital too. Schools are ideal environments to encourage wellness in the future generations because they provide opportunities for students to learn and practice diverse healthy behaviors and eating habits. Apart from academic lessons, students also learn from social and emotional relationships with their peers and teachers, and this may, in turn, affect their health behaviors; this is especially significant given that school-age adolescents spend more than half of their days at school, and schools provide about 50% of children’s daily caloric intake [45].

This study should be interpreted in light of the following limitations. First, this study utilized self-reported measurements to assess all study variables through students’ one-day recollections, which may be inaccurate due to recall bias, respondent bias, or interview bias. Second, the small sample size (e.g., Hispanic and Asian) and limited sampling region warn against the generalization of the research findings to those living in other more diversely populated areas. Studies conducted with larger and more diverse samples could lead to better results. Third, this study was conducted in Southeast US. Therefore, its results cannot be generalized to adolescents in other states or countries. Finally, the YRBSS dataset excludes households without telephones, which may result in a biased survey population, due to the underrepresentation of certain segments of the population. Therefore, future evaluation and research studies regarding children’s obesity prevention should take these limitations into consideration. However, we believe that these limitations do not outweigh the contribution of this study.

## 5. Conclusions

Despite the huge benefits associated with the prevention of childhood obesity, preventable obesity intervention and service programs are not fully provided for adolescents [46]. A commitment to and implementation of intervention programs or policies for childhood obesity are vital for health educators and professionals, policy makers, and stakeholders in improving the quality of life of children and adolescents. One of the biggest reasons policy-makers or stakeholders should implement health-related policies or interventions is that the cost–benefit ratio of medical or social interventions demonstrates that the highest developmental stage/approximate is in childhood [47]. Therefore, if policy-makers desire better results, the developmental stage (e.g., childhood and adolescence) should be targeted, in terms of the cost-benefit ratio. In other words, smaller rewards are expected with respect to medical or social interventions, in adulthood.

In the past decade, several states have implemented legislation concerning school polices for childhood obesity, focusing on nutrition standards, nutrition education, and physical activity. The evidence of this study supports the effectiveness of such policies in Mississippi, and describes how school-based intervention policies affect students’ practice and the implementation of obesity prevention behaviors. School-based nutrition and physical activity policies discourage both children and adolescents from engaging in abusive behaviors to lose weight, encourage students’ consumption of fruits and vegetables and participation in physical activity, and decrease the amount of watching TV. As illustrated in this study, school polices related to children’s eating and physical activity behaviors could have a huge impact on reducing childhood obesity.

## Figures and Tables

**Figure 1 ijerph-17-09425-f001:**
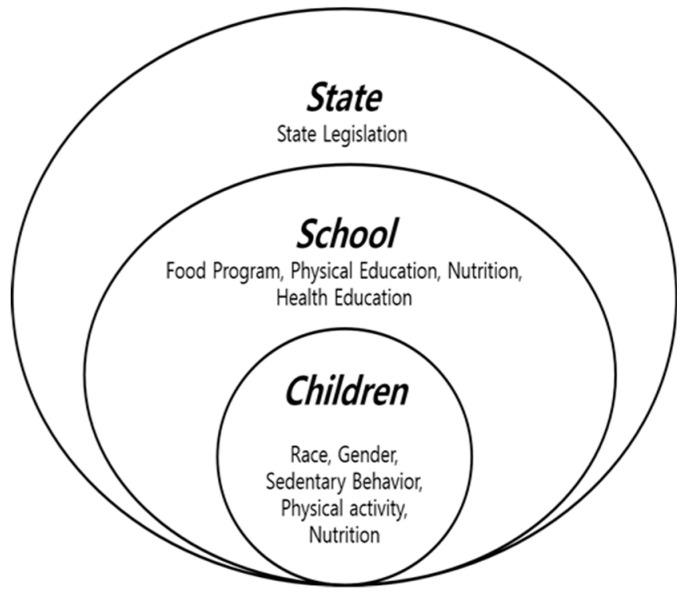
The adopted ecological system theory from Bronfenbrenner (1979, 1989).

**Figure 2 ijerph-17-09425-f002:**
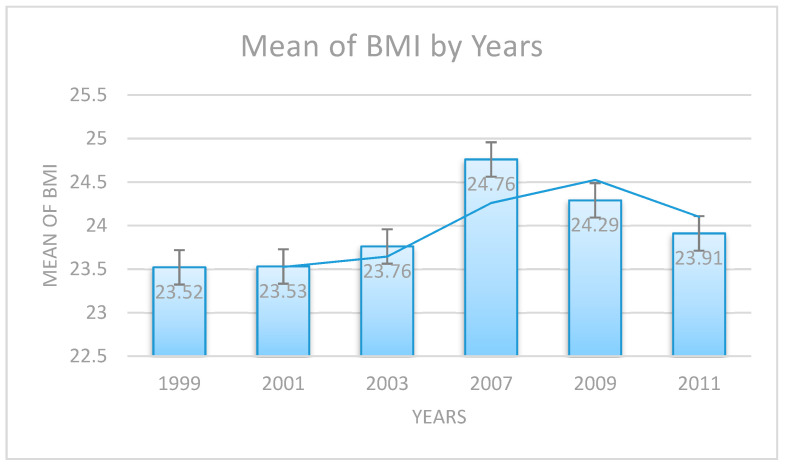
The age-, sex-, and ethnicity-adjusted mean of the body mass index by years.

**Figure 3 ijerph-17-09425-f003:**
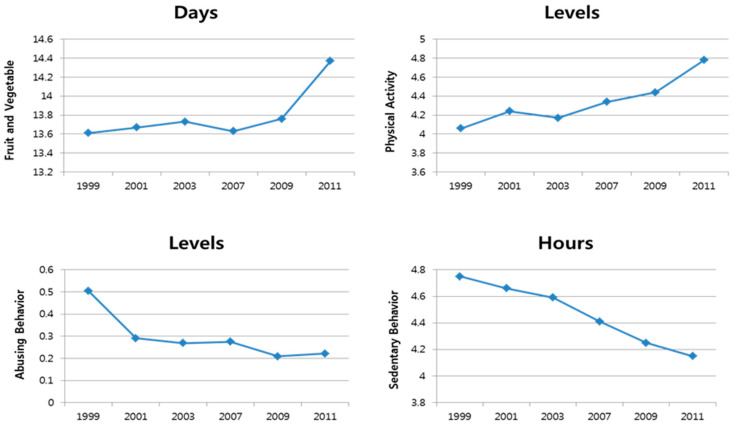
Variable scores by years.

**Table 1 ijerph-17-09425-t001:** Description of the participants and their nutrition- and physical activity-related behaviors.

Age	16.21 Years (SD = 1.23)	Variables				
	Overall % (N)	Body Mass Index	Consumption of Fruits and Vegetables	Sedentary Behavior	Physical Activity	Abusive Behavior to Lose Weight
Sex:		*t*_(8859)_ −7.41 **	*t*_(8860)_ −6.79 **	*t*_(8844)_ −1.80	*t*_(8860)_ −23.84 **	*t*_(8860)_ −13.52 **
Male	46.2% (4095)					
Female	53.8% (4767)					
Ethnicity:		*F*_(2, 8785)_ 90.34 **	*F*_(2, 8786)_ 26.92 **	*F*_(2, 8772)_ 491.16 **	*F*_(2, 8776)_ 36.55 **	*F*_(2, 8786)_ 15.48 **
White	46.5% (4118)					
Black	46.7% (4137)					
Others	6.0% (534)					
Education:		*F*_(4, 8848)_ 12.35 **	*F*_(4, 8845)_ 1.98	*F*_(4, 8829)_ 8.82 **	*F*_(4, 8845)_ 69.65 **	*F*_(4, 8845)_ 0.95
9th grade	29.0% (2573)					
10th grade	21.7% (1922)					
11th grade	24.1% (2133)					
12th grade	24.5% (2169)					

** *p* < 0.01. Note: Significance was determined by covariate analysis. Data source: 1999–2011 Youth Risk Behavior Surveillance System in the State of Mississippi.

**Table 2 ijerph-17-09425-t002:** Differences in the variables of the years, from 1999 to 2011, from the Factorial ANOVA.

Variables		Year						*p*-Value
		1999	2001	2003	2007	2009	2011	
Consumption of fruits and vegetables	*N*	1355	1501	1234	1425	1638	1709	*p* < 0.01 **
	*M*	13.61	13.67	13.73	13.63	13.76	14.37	
	*SD*	5.321	5.044	5.175	5.644	5.384	5.758	
Sedentary behavior	*N*	1354	1500	1231	1423	1633	1705	*p* < 0.01 **
	*M*	4.75	4.66	4.59	4.41	4.25	4.15	
	*SD*	1.825	1.888	1.791	1.945	1.888	1.587	
Physical activity	*N*	1355	1501	1234	1425	1638	1709	*p* < 0.01 **
	*M*	4.06	4.24	4.17	4.34	4.44	4.78	
	*SD*	2.540	2.697	2.676	2.652	2.751	2.715	
Abusive behavior to lose weight	*N*	1355	1501	1234	1425	1638	1709	*p* < 0.01 **
	*M*	0.504	0.291	0.269	0.275	0.209	0.221	
	*SD*	1.122	0.653	0.628	0.596	0.535	0.555	

*N* = number of variable, *M* = mean, *SD* = standard deviation. ** *p* < 0.01. Data source: 1999–2011 Youth Risk Behavior Surveillance System in the State of Mississippi.

**Table 3 ijerph-17-09425-t003:** Multivariate logistic regression to identify the factors associated with students’ overweight status.

Variable	Adjusted Odds Ratios	95% Confidence Interval	*p*-Value
Consumption of fruits and vegetables	1.00	(1.00, 1.01)	0.226
Watching TV	1.05	(1.02, 1.07)	0.001 **
Physical activity	0.97	(0.96, 0.99)	0.003 **
Abusive behavior to lose weight	1.37	(1.29, 1.46)	<0.001 **
Age	0.92	(0.88, 0.95)	<0.001 **
Ethnicity			
White			
Black	0.642	(0.582, 0.709)	<0.001 **
Other	0.942	(0.774, 1.147)	0.553
Sex			
Male			
Female	1.585	(1.443, 1.742)	<0.001 **
Year			
1999			
2001	0.951	(0.809, 1.119)	0.548
2003	0.853	(0.720, 1.010)	0.065
2007	0.686	(0.583, 0.807)	<0.001 **
2009	0.739	(0.631, 0.867)	<0.001 **
2011	0.760	(0.649, 0.891)	0.001 **

** *p* < 0.01. Data source: 1999–2011 Youth Risk Behavior Surveillance System in the State of Mississippi.

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
