# Peer review of "Multi-Year Examination of School-Based Programs in Preventing Childhood Obesity: A Case of a State Policy in the U.S."

_ijerph, 2020, doi:10.3390/ijerph17249425_

Round 1

Reviewer 1 Report

Dear authors,

Below you can find my comments on your paper entitled Multi-Year Examination of School-based Programs in Preventing Childhood Obesity: A Case of a State Policy in the U.S.

General comment

The article is largely well written but I have serious doubts about the use of data available on websites Youth Risk Behavior Survey in the State of Mississippi (https://msdh.ms.gov/msdhsite/_static/31,0,302.html) and Youth Risk Behavior Surveillance System (https://www.cdc.gov/healthyyouth/data/yrbs/index.htm) in this article. Did the authors have anything to do with this research? Using data from such a project and calculating the results only on the basis of this data suggest participation in this project. The situation would be different if it were a review article gathering data from different studies or comparing the effectiveness of different interventions. Moreover, the research results are published on the above-mentioned websites and are available, so does it make sense to publish this article in its form?

Detailed comments:

  • The research concerned adolescents, however, Key words include the “childhood”, and some parts of Introduction and Discussion also concern the childhood (e.g. lines 29, 36, 45, 49, 240).
  • I propose to describe the changes introduced into the school-based policies of the State in detail.
  • Poorly described sample selection and research method.
  • In the description of the respondents, it was not mentioned that only data from the State of Mississippi was used.
  • What was the age range?
  • The measurement of physical activity should be clarified.
  • Sedentary behavior - only watching TV was considered?
  • There is no “n/s” notation in table 2, which is in the table footnote.
  • The results should be clarified - some of them are not in the table (lines 164-166). The description is unclear (lines 164-176).
  • Not all tables have a caption where the data was taken from.
  • Standardize the spacing between lines of the text.
  • Authors Contributions and Funding should be completed.

Author Response

Dear Reviewer, 

We are very grateful for your valuable and meaningful comments which are very helpful to our paper. We tried our best to conform to your comments. The following is a one-on-one answer to your comment. Please see the attachment.

Thank you again for your hard work.

Best warmly, 

Point 1: The article is largely well written but I have serious doubts about the use of data available on websites Youth Risk Behavior Survey in the State of Mississippi (https://msdh.ms.gov/msdhsite/_static/31,0,302.html) and Youth Risk Behavior Surveillance System (https://www.cdc.gov/healthyyouth/data/yrbs/index.htm) in this article. Did the authors have anything to do with this research? Using data from such a project and calculating the results only on the basis of this data suggest participation in this project. The situation would be different if it were a review article gathering data from different studies or comparing the effectiveness of different interventions. Moreover, the research results are published on the above-mentioned websites and are available, so does it make sense to publish this article in its form?

Response 1: Thank you for your valuable and important comments and comments. Overall Youth Risk Behavior Surveillance System (https://www.cdc.gov/healthyyouth/data/yrbs/index.htm) data is publicly available at the above site. However, this data is a combination of data from all states in the United States, and data for specific individual states are not available. So, the author of this paper contacted the Mississippi CDC to use the data from the Youth Risk Behavior Survey in the State of Mississippi (https://msdh.ms.gov/msdhsite/_static/31,0,302.html). It took me one month to fill out the data request form. This data is more specific and more detailed than the Youth Risk Behavior Surveillance System (https://www.cdc.gov/healthyyouth/data/yrbs/index.htm) data.

Detailed comments:

Point 2: The research concerned adolescents, however, Keywords include the “childhood”, and some parts of Introduction and Discussion also concern the childhood (e.g. lines 29, 36, 45, 49, 240).

Response 2: Participants in this study are children and adolescents aged 14 to 18 years old from 9th to 12th grade. Therefore, in a slightly broader sense, the expression was used as childhood.

Point 3: I propose to describe the changes introduced into the school-based policies of the State in detail.

Response 3: We described it in lines 51 to 61. Both Mississippi and Louisiana states have the same policy but the time implemented it was different.

Point 4: Poorly described sample selection and research method.

Response 4: We reinforced line of 97 for data sample selection and lines of 101 to 103 study method with red.

Point 5: In the description of the respondents, it was not mentioned that only data from the State of Mississippi was used.

Response 5: We put it in the line of 97 with red.

Point 6: What was the age range?

Response 6: This study of participants’ age range was 14 to 18.

Point 7: The measurement of physical activity should be clarified.

Response 7: We put more details of physical activity measures in the line of 126 to 127 with red.

Point 8: Sedentary behavior - only watching TV was considered?

Response 8: Yes. In the data collection method up to 2011, only TV viewing time was used to measure sedentary levels.

Point 9: There is no “n/s” notation in table 2, which is in the table footnote.

Response 9: We deleted it.

Point 10: The results should be clarified - some of them are not in the table (lines 164-166). The description is unclear (lines 164-176).

Response 10: We made it correct to be more reader-able in the line of 171 to 180 with red

Point 11: Not all tables have a caption where the data was taken from.

Response 11: We put all captions where the data come from.

Point 12: Standardize the spacing between lines of the text.

Response 12: We corrected it.

Point 13: Authors Contributions and Funding should be completed.

Response 13: We corrected and revised it in the line of 337 to 342.

Reviewer 2 Report

Dear Authors,

The work you present is of great interest to the journal readers, especially to policymakers. The introduction summarizes well the context of the study, the need to implement such policies and the need to evaluate them.

However, I believe that the methodology and results proposed do not respond to the objective of the study, which is to test the effectiveness of school policies for obesity prevention. First of all, the prevalence or incidence of overweight/obesity in the sample is not provided, only the BMI as a continuous variable. The BMI at school age grows naturally, so the use of percentiles or z-score with respect to the reference is more pertinent than crude BMI (if each year the composition of the sample varies in average age, this may have a direct effect on the average BMI and that is not indicative of improvement/ worsening of overweight or obesity over time).

On the other hand, most of the results are aimed at testing the association between the different variables in the study and the risk of suffering from weight excess. But no direct comparison is made between the results of these variables or of the relationship of these variables to obesity before and after the implementation of the policies (as expected after reading the introduction). In general, you make statements and conclusions that are not supported by the analyses and results included in the manuscript. 

Another important point is that you provide p values for the comparison of a variable between two groups, but do not provide the average values in these two groups, so although it is shown that there are differences, it is not possible to know in what direction they are going or how to interpret them (in which group is higher/lower? do they improve? do they get worse?).

Nevertheless, the results seem to show an improvement in terms of childhood habits (especially for fruit and vegetable consumption). Despite the weaknesses mentioned, I consider the study to be of great interest and relevance, and the results interesting so I think it could be potentially published if a thorough review of the manuscript is done. I am enclosing it with more detailed comments that I believe would help to improve the quality of the work.

Best regards,

Author Response

Dear Reviewer,

We are very grateful for your valuable and meaningful comments that are very helpful to our paper. We've done our best to match your comments. Here is a one-to-one answer to your comment. Please see the attachment.

Thank you again for your hard work.

Best Warmly, 

Point 1: The work you present is of great interest to the journal readers, especially to policymakers. The introduction summarizes well the context of the study, the need to implement such policies and the need to evaluate them.

However, I believe that the methodology and results proposed do not respond to the objective of the study, which is to test the effectiveness of school policies for obesity prevention. First of all, the prevalence or incidence of overweight/obesity in the sample is not provided, only the BMI as a continuous variable. The BMI at school age grows naturally, so the use of percentiles or z-score with respect to the reference is more pertinent than crude BMI (if each year the composition of the sample varies in average age, this may have a direct effect on the average BMI and that is not indicative of improvement/ worsening of overweight or obesity over time).

Response 1: To check the effectiveness of school obesity policy, from 1999 to 2011 the age-, sex-, and ethnicity-adjusted mean of the body mass index were identified, and the average BMI of children decreased around 2007 when the policy was timed in the line of 152 to 154 with red.

Point 2: On the other hand, most of the results are aimed at testing the association between the different variables in the study and the risk of suffering from weight excess. But no direct comparison is made between the results of these variables or of the relationship of these variables to obesity before and after the implementation of the policies (as expected after reading the introduction). In general, you make statements and conclusions that are not supported by the analyses and results included in the manuscript. 

Response 2: How the variables of physical activity and nutrition, which are significantly related to obesity, increased and decreased from 1999 to 2011 at the beginning of the study, showed how policies affected children's weight changes.

Point 3: Another important point is that you provide p values for the comparison of a variable between two groups, but do not provide the average values in these two groups, so although it is shown that there are differences, it is not possible to know in what direction they are going or how to interpret them (in which group is higher/lower? do they improve? do they get worse?).

Response 3: The purpose of this study was to assess the role of the state policy for schools in Mississippi in the United States on child obesity, not to analyze each child's obesity. Therefore, no direct comparison was made between obese children and those who did not.

Round 2

Reviewer 1 Report

Thanks to the authors for taking into account the suggestions and making changes to the manuscript. I believe that this manuscript could be accepted for publication.

Author Response

Dear Reviewer,

I really appreciate your valuable comments.

Best regards,

TaeEung Kim, Ph.D.